# Understanding Posterior Collapse in Generative Latent Variable Models

**James Lucas[‡*], George Tucker[†], Roger Grosse[‡], Mohammad Norouzi[†]**
†Google Brain, ‡University of Toronto

## Abstract

Posterior collapse in Variational Autoencoders (VAEs) arises when the variational distribution closely matches the uninformative prior for a subset of latent variables. This paper presents a simple and intuitive explanation for posterior collapse through the analysis of linear VAEs and their direct correspondence with Probabilistic PCA (pPCA). We identify how local maxima can emerge from the marginal log-likelihood of pPCA, which yields similar local maxima for the evidence lower bound (ELBO). We show that training a linear VAE with variational inference recovers a uniquely identifiable global maximum corresponding to the principal component directions. We provide empirical evidence that the presence of local maxima causes posterior collapse in non-linear VAEs. Our findings help to explain a wide range of heuristic approaches in the literature that attempt to diminish the effect of the KL term in the ELBO to alleviate posterior collapse.

## 1 Introduction

The generative process of a deep latent variable model entails drawing a number of latent factors from an uninformative prior and using a neural network to convert such factors to real data points. Maximum likelihood estimation of the parameters requires marginalizing out the latent factors, which is intractable for deep latent variable models. The influential work of Kingma & Welling (2013) and Rezende et al. (2014) on Variational Autoencoders (VAEs) enables optimization of a tractable lower bound on the likelihood via a reparameterization of the Evidence Lower Bound (ELBO) (Jordan et al., 1999; Blei et al., 2017). This has created a surge of recent interest in automatic discovery of the latent factors of variation for a data distribution based on VAEs and principled probabilistic modeling (Higgins et al., 2016; Bowman et al., 2015; Chen et al., 2018; Gomez-Bombarelli et al., 2018).

Unfortunately, the quality and the number of the latent factors learned is directly controlled by the extent of a phenomenon known as *posterior collapse*, where the generative model learns to ignore a subset of the latent variables. Most existing work suggests that posterior collapse is caused by the KL-divergence term in the ELBO objective, which directly encourages the variational distribution to match the prior. Thus, a wide range of heuristic approaches in the literature have attempted to diminish the effect of the KL term in the ELBO to alleviate posterior collapse. By contrast, we hypothesize that posterior collapse arises due to spurious local maxima in the training objective. Surprisingly, we show that these local maxima may arise even when training with exact marginal log-likelihood.

While linear autoencoders (Rumelhart et al., 1985) have been studied extensively (Baldi & Hornik, 1989; Kunin et al., 2019), little attention has been given to their variational counterpart. A well-known relationship exists between linear autoencoders and PCA — the optimal solution to the linear autoencoder problem has decoder weight columns which span the subspace defined by the principal components. The Probabilistic PCA (pPCA) model (Tipping & Bishop, 1999) recovers the principal component subspace as the maximum likelihood solution of a Gaussian latent variable model. In this work, we show that pPCA is recovered exactly using linear variational autoencoders. Moreover, by specifying a diagonal

---

*Intern at Google Brain

covariance structure on the variational distribution we recover an identifiable model which at the global maximum has the principal components as the columns of the decoder.

The study of linear VAEs gives us new insights into the cause of posterior collapse. Following the analysis of Tipping & Bishop (1999), we characterize the stationary points of pPCA and show that the *variance of the observation model* directly impacts the stability of local stationary points – if the variance is too large then the pPCA objective has spurious local maxima, which correspond to a collapsed posterior. Our contributions include:

- We prove that linear VAEs can recover the true posterior of pPCA and using ELBO to train linear VAEs does not add any additional spurious local maxima. Further, we prove that at its global optimum, the linear VAE recovers the principal components.
- We shows that posterior collapse may occur in optimization of marginal log-likelihood, without powerful decoders. Our experiments verify the analysis of the linear setting and show that these insights extend even to high-capacity, deep, non-linear VAEs.
- By learning the observation noise carefully, we are able to reduce posterior collapse. We present evidence that the success of existing approaches in alleviating posterior collapse depends on their ability to reduce the stability of spurious local maxima.

## 2 Preliminaries

**Probabilistic PCA.** We define the probabilitic PCA (pPCA) model as follows. Suppose latent variables $\mathbf{z} \in \mathbb{R}^k$ generate data $\mathbf{x} \in \mathbb{R}^n$. A standard Gaussian prior is used for $\mathbf{z}$ and a linear generative model with a spherical Gaussian observation model for $\mathbf{x}$:

$$
\begin{aligned}
p(\mathbf{z}) &= \mathcal{N}(\mathbf{0}, \mathbf{I}) \\
p(\mathbf{x} \mid \mathbf{z}) &= \mathcal{N}(\mathbf{W}\mathbf{z} + \boldsymbol{\mu}, \sigma^2 \mathbf{I})
\end{aligned}
\tag{1}
$$

The pPCA model is a special case of factor analysis (Bartholomew, 1987), which replaces the spherical covariance $\sigma^2 \mathbf{I}$ with a full covariance matrix. As pPCA is fully Gaussian, both the marginal distribution for $\mathbf{x}$ and the posterior $p(\mathbf{z}|\mathbf{x})$ are Gaussian and, unlike factor analysis, the maximum likelihood estimates of $\mathbf{W}$ and $\sigma^2$ are tractable (Tipping & Bishop, 1999).

**Variational Autoencoders.** Recently, amortized variational inference has gained popularity as a means to learn complicated latent variable models. In these models, the marginal log-likelihood, $\log p(\mathbf{x})$, is intractable but a variational distribution, $q(\mathbf{z}|\mathbf{x})$, is used to approximate the posterior, $p(\mathbf{z}|\mathbf{x})$, allowing tractable approximate inference. To do so we typically make use of the Evidence Lower Bound (ELBO):

$$
\begin{aligned}
\log p(\mathbf{x}) &= \mathbb{E}_{q(\mathbf{z}|\mathbf{x})}[\log p(\mathbf{x}, \mathbf{z}) - \log q(\mathbf{z} \mid \mathbf{x})] + D_{KL}(q(\mathbf{z} \mid \mathbf{x}) || p(\mathbf{z} \mid \mathbf{x})) &&(2) \\
&\geq \mathbb{E}_{q(\mathbf{z}|\mathbf{x})}[\log p(\mathbf{x}, \mathbf{z}) - \log q(\mathbf{z} \mid \mathbf{x})] &&(3) \\
&= \mathbb{E}_{q(\mathbf{z}|\mathbf{x})}[\log p(\mathbf{x} \mid \mathbf{z})] - D_{KL}(q(\mathbf{z} \mid \mathbf{x}) || p(\mathbf{z})) \qquad (:= ELBO) &&(4)
\end{aligned}
$$

The ELBO consists of two terms, the KL divergence between the variational distribution, $q(\mathbf{z}|\mathbf{x})$, and prior, $p(\mathbf{z})$, and the expected conditional log-likelihood. The KL divergence forces the variational distribution towards the prior and so has reasonably been the focus of many attempts to alleviate posterior collapse. We hypothesize that in fact the marginal log-likelihood itself often encourages posterior collapse.

In Variational Autoencoders (VAEs), two neural networks are used to parameterize $q_\phi(\mathbf{z}|\mathbf{x})$ and $p_\theta(\mathbf{x}|\mathbf{z})$, where $\phi$ and $\theta$ denote two sets of neural network weights. The encoder maps an input $\mathbf{x}$ to the parameters of the variational distribution, and then the decoder maps a sample from the variational distribution back to the inputs.

**Posterior collapse.** The most consistent issue with VAE optimization is posterior collapse, in which the variational distribution collapses towards the prior: $\exists i \ s.t. \ \forall \mathbf{x} \ q_\phi(z_i|\mathbf{x}) \approx p(z_i)$. This reduces the capacity of the generative model, making it impossible for the decoder network to make use of the information content of all of the latent dimensions. While posterior collapse is typically described using the variational distribution as above, one can also define it in terms of the true posterior $p(z|\mathbf{x})$ as: $\exists i \ s.t. \ \forall \mathbf{x} \ p(z_i|\mathbf{x}) \approx p(z_i)$.

## 3 RELATED WORK

Dai et al. (2017) discuss the relationship between robust PCA methods (Candès et al., 2011) and VAEs. In particular, they show that at stationary points the VAE objective locally aligns with pPCA under certain assumptions. We study the pPCA objective explicitly and show a direct correspondence with linear VAEs. Dai et al. (2017) show that the covariance structure of the variational distribution may help smooth out the loss landscape. This is an interesting result whose interactions with ours is an exciting direction for future research.

He et al. (2019) motivate posterior collapse through an investigation of the learning dynamics of deep VAEs. They suggest that posterior collapse is caused by the inference network lagging behind the true posterior during the early stages of training. A related line of research studies issues arising from approximate inference causing mismatch between the variational distribution and true posterior (Cremer et al., 2018; Kim et al., 2018; Hjelm et al., 2016). By contrast, we show that local maxima may exist even when the variational distribution matches the true posterior exactly.

Alemi et al. (2017) use an information theoretic framework to study the representational properties of VAEs. They show that with infinite model capacity there are solutions with equal ELBO and marginal log-likelihood which span a range of representations, including posterior collapse. We find that even with weak (linear) decoders, posterior collapse may occur. Moreover, we show that in the linear case this posterior collapse is due entirely to the marginal log-likelihood.

The most common approach for dealing with posterior collapse is to anneal a weight on the KL term during training from 0 to 1 (Bowman et al., 2015; Sønderby et al., 2016; Maaløe et al., 2019; Higgins et al., 2016; Huang et al., 2018). Unfortunately, this means that during the annealing process, one is no longer optimizing a bound on the log-likelihood. In addition, it is difficult to design these annealing schedules and we have found that once regular ELBO training resumes the posterior will typically collapse again (Section 5.2).

Kingma et al. (2016) propose a constraint on the KL term, which they called "free-bits", where the gradient of the KL term per dimension is ignored if the KL is below a given threshold. Unfortunately, this method reportedly has some negative effects on training stability (Razavi et al., 2019; Chen et al., 2016). Delta-VAEs (Razavi et al., 2019) instead choose prior and variational distributions carefully such that the variational distribution can never exactly recover the prior, allocating free-bits implicitly.

Several other papers have studied alternative formulations of the VAE objective (Rezende & Viola, 2018; Dai & Wipf, 2019; Alemi et al., 2017). Dai & Wipf (2019) analyze the VAE objective with the goal of improving image fidelity under Gaussian observation models. Through this lens they discuss the importance of the observation noise.

Rolinek et al. (2018) point out that due to the diagonal covariance used in the variational distribution of VAEs they are encouraged to pursue orthogonal representations. They use linearizations of deep networks to prove their results under a modification of the objective function by explicitly ignoring latent dimensions with posterior collapse. Our formulation is distinct in focusing on linear VAEs without modifying the objective function and proving an exact correspondence between the global solution of linear VAEs and principal components.

Kunin et al. (2019) studies the optimization challenges in the linear autoencoder setting. They expose an equivalence between pPCA and Bayesian autoencoders and point out that when $\sigma^2$ is too large information about the latent code is lost. A similar phenomenon is discussed in the supervised learning setting by Chechik et al. (2005). Kunin et al. (2019) also show that suitable regularization allows the linear autoencoder to exactly recover the principal components. We show that the same can be achieved using linear variational autoencoders with a diagonal covariance structure.

## 4 ANALYSIS OF LINEAR VAE

In this section we compare and analyze the optimal solutions to both pPCA and linear variational autoencoders.

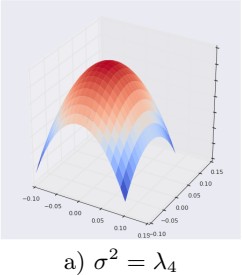 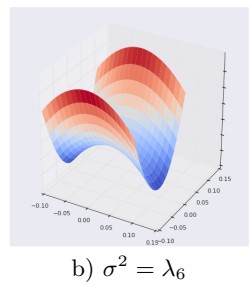 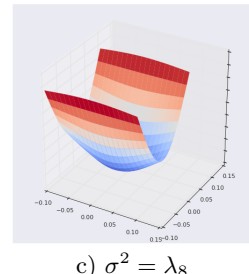

a) $\sigma^2 = \lambda_4$        b) $\sigma^2 = \lambda_6$        c) $\sigma^2 = \lambda_8$

Figure 1: **Stationary points of pPCA.** A zero-column of $\mathbf{W}$ is perturbed in the directions of two orthogonal principal components ($\mu_5$ and $\mu_7$) and the loss surface (marginal log-likelihood) is shown. The stability of the stationary points depends critically on $\sigma^2$. Left: $\sigma^2$ is able to capture both principal components. Middle: $\sigma^2$ is too large to capture one of the principal components. Right: $\sigma^2$ is too large to capture either principal component.

We first discuss the maximum likelihood estimates of pPCA and then show that a simple linear VAE is able to recover the global optimum. Moreover, the same linear VAE recovers identifiability of the principle components (unlike pPCA which only spans the PCA subspace). Finally, we analyze the loss landscape of the linear VAE showing that ELBO does not introduce any additional spurious maxima.

### 4.1 PROBABILISTIC PCA REVISITED

The pPCA model (Eq. (1)) is a fully Gaussian linear model and thus we can compute both the marginal distribution for $\mathbf{x}$ and the posterior $p(\mathbf{z} \mid \mathbf{x})$ in closed form:

$$p(\mathbf{x}) = \mathcal{N}(\boldsymbol{\mu}, \mathbf{W}\mathbf{W}^T + \sigma^2 \mathbf{I}), \tag{5}$$

$$p(\mathbf{z} \mid \mathbf{x}) = \mathcal{N}(\mathbf{M}^{-1}\mathbf{W}^T(\mathbf{x} - \boldsymbol{\mu}), \sigma^2 \mathbf{M}^{-1}), \tag{6}$$

where $\mathbf{M} = \mathbf{W}^T\mathbf{W} + \sigma^2 \mathbf{I}$. This model is particularly interesting to analyze in the setting of variational inference as the ELBO can also be computed in closed form (see Appendix C).

**Stationary points of pPCA**    We now characterize the stationary points of pPCA, largely repeating the thorough analysis of Tipping & Bishop (1999) (see Appendix A of their paper).

The maximum likelihood estimate of $\boldsymbol{\mu}$ is the mean of the data. We can compute $\mathbf{W}_{\text{MLE}}$ and $\sigma_{\text{MLE}}$ as follows:

$$\sigma^2_{\text{MLE}} = \frac{1}{n-k} \sum_{j=k+1}^{n} \lambda_j, \tag{7}$$

$$\mathbf{W}_{\text{MLE}} = \mathbf{U}_k(\boldsymbol{\Lambda}_k - \sigma^2_{\text{MLE}}\mathbf{I})^{1/2}\mathbf{R}. \tag{8}$$

Here $\mathbf{U}_k$ corresponds to the first $k$ principal components of the data with the corresponding eigenvalues $\lambda_1, \ldots, \lambda_k$ stored in the $k \times k$ diagonal matrix $\boldsymbol{\Lambda}_k$. The matrix $\mathbf{R}$ is an arbitrary rotation matrix which accounts for weak identifiability in the model. We can interpret $\sigma^2_{MLE}$ as the average variance lost in the projection. The MLE solution is the global optima.

**Stability of $\mathbf{W}_{\text{MLE}}$**    One surprising observation is that $\sigma^2$ directly controls the stability of the stationary points of the marginal log-likelihood (see Appendix A). In Figure 1, we illustrate one such stationary point of pPCA under different values of $\sigma^2$. We computed this stationary point by taking $\mathbf{W}$ to have three principal components columns and zeros elsewhere. Each plot shows the same stationary point perturbed by two orthogonal eigenvectors corresponding to other principal components. The stability of the stationary points depends on the size of $\sigma^2$ — as $\sigma^2$ increases the stationary point tends towards a stable local maxima. While this example is much simpler than a non-linear VAE, we find in practice that the same principle applies. Moreover, we observed that the non-linear dynamics make it difficult to learn a smaller value of $\sigma^2$ automatically (Figure 6).

## 4.2 Linear VAEs recover pPCA

We now show that linear VAEs are able to recover the globally optimal solution to Probabilistic PCA. We will consider the following VAE model,

$$
\begin{aligned}
p(\mathbf{x} \mid \mathbf{z}) &= \mathcal{N}(\mathbf{W}\mathbf{z} + \boldsymbol{\mu}, \sigma^2 \mathbf{I}), \\
q(\mathbf{z} \mid \mathbf{x}) &= \mathcal{N}(\mathbf{V}(\mathbf{x} - \boldsymbol{\mu}), \mathbf{D}),
\end{aligned}
\tag{9}
$$

where $\mathbf{D}$ is a diagonal covariance matrix which is used globally for all data points. While this is a significant restriction compared to typical VAE architectures, which define an amortized variance for each input point, this is sufficient to recover the global optimum of the probabilistic model.

**Lemma 1.** *The global maximum of the ELBO objective (Eq. (4)) for the linear VAE (Eq. (9)) is identical to the global maximum for the marginal log-likelihood of pPCA (Eq. (5)).*

*Proof.* The global optimum of pPCA is obtained at the maximum likelihood estimate of $\mathbf{W}$ and $\sigma^2$, which are specified only up to an orthogonal transformation of the columns of $\mathbf{W}$, *i.e.,* any rotation $\mathbf{R}$ in Eq. (8) results in a matrix $\mathbf{W}_{\text{MLE}}$ that given $\sigma^2_{\text{MLE}}$ attains maximum marginal likelihood. The linear VAE model defined in Eq. (9) is able to recover the global optimum of pPCA only when $\mathbf{R} = \mathbf{I}$. When $\mathbf{R} = \mathbf{I}$, we have $\mathbf{M} = \mathbf{W}^T_{\text{MLE}} \mathbf{W}_{\text{MLE}} + \sigma^2 \mathbf{I} = \boldsymbol{\Lambda}_k$, thus setting $\mathbf{V} = \mathbf{M}^{-1} \mathbf{W}^T_{\text{MLE}}$ and $\mathbf{D} = \sigma^2_{\text{MLE}} \mathbf{M}^{-1} = \sigma^2_{\text{MLE}} \boldsymbol{\Lambda}_k^{-1}$ (which is diagonal) recovers the true posterior at the global optimum. In this case, the ELBO equals the marginal log-likelihood and is maximized when the decoder has weights $\mathbf{W} = \mathbf{W}_{\text{MLE}}$. Since, ELBO lower bounds log-likelihood, then the global maximum of ELBO for the linear VAE is the same as the global maximum of marginal likelihood for pPCA. $\square$

Full details are given in Appendix C. In fact, the diagonal covariance of the variational distribution allows us to identify the principal components at the global optimum.

**Corollary 1.** *The global optimum to the VAE solution has the scaled principal components as the columns of the decoder network.*

*Proof.* Follows directly from the proof of Lemma 1 and Equation 8. $\square$

Finally, we can recover full identifiability by requiring $\mathbf{D} = \mathbf{I}$. We discuss this in Appendix B.

We have shown that at its global optimum the linear VAE is able to recover the pPCA solution and additionally enforces orthogonality of the decoder weight columns. However, the VAE is trained with the ELBO rather than the marginal log-likelihood. The majority of existing work suggests that the KL term in the ELBO objective is responsible for posterior collapse and so we should ask whether this term introduces additional spurious local maxima. Surprisingly, for the linear VAE model the ELBO objective *does not* introduce any additional spurious local maxima. We provide a sketch of the proof here with full details in Appendix C.

**Theorem 1.** *The ELBO objective does not introduce any additional local maxima to the pPCA model.*

*Proof.* (Sketch) If the decoder network has orthogonal columns then the variational distribution can capture the true posterior and thus the variational objective exactly recovers the marginal log-likelihood at stationary points. If the decoder network does not have orthogonal columns then the variational distribution is no longer tight. However, the ELBO can always be increased by rotating the columns of the decoder towards orthogonality. This is because the variational distribution fits the true posterior more closely while the marginal log-likelihood is invariant to rotations of the weight columns. Thus, any additional stationary points in the ELBO objective must necessarily be saddle points. $\square$

The theoretical results presented in this section provide new intuition for posterior collapse in general VAEs. Our results suggest that the ELBO objective, in particular the KL between the variational distribution and the prior, is not entirely responsible for posterior collapse — even exact marginal log-likelihood may suffer. The evidence for this is two-fold. We have shown that marginal log-likelihood may have spurious local maxima but also that in the linear case the ELBO objective does not add any additional spurious local maxima. Rephrased, in the linear setting the problem lies entirely with the probabilistic model.

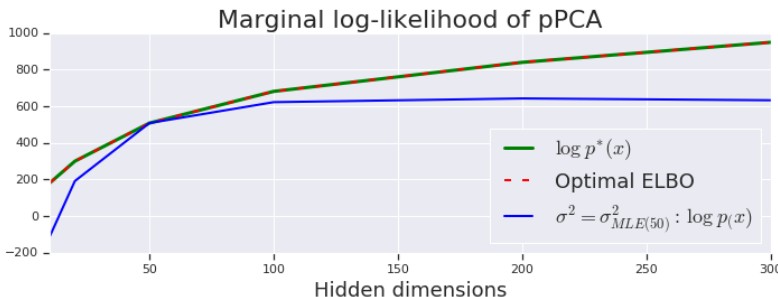

Figure 2: The marginal log-likelihood and optimal ELBO of MNIST pPCA solutions over increasing hidden dimension. Green represents the MLE solution (global maximum), the red dashed line is the optimal ELBO solution which matches the global optimum. The blue line shows the marginal log-likelihood of the solutions using the full decoder weights when $\sigma^2$ is fixed to its MLE solution for 50 hidden dimensions.

## 5  EXPERIMENTS

In this section we present empirical evidence found from studying two distinct claims. First, we verified our theoretical analysis of the linear VAE model. Second, we explored to what extent these insights apply to deep non-linear VAEs.

### 5.1  LINEAR VAEs

In Figure 2 we display the likelihood values for various optimal solutions to the pPCA model trained on the MNIST dataset. We plot the maximum log-likelihood and numerically verify that the optimal ELBO solution is able to exactly match this (Lemma 1). We also evaluated the model with all principal components used but with a fixed value of $\sigma^2$ corresponding to the MLE solution for 50 hidden dimensions. This is equivalent to $\sigma^2 \approx \lambda_{222}$. Here the log-likelihood is optimal at $\sigma^2 = 50$ as expected, but interestingly the likelihood decreases for 300 hidden dimensions — including the additional principal components has made the solution worse under marginal log-likelihood.

### 5.2  INVESTIGATING POSTERIOR COLLAPSE IN DEEP NON-LINEAR VAEs

We explored how well the analysis of the linear VAEs extends to deep non-linear models. To do so, we trained VAEs with Gaussian observation models on the MNIST dataset. This is a fairly uncommon choice of model for this dataset, which is nearly binary, but it provides a good setting for us to investigate our theoretical findings.

**Training with fixed $\sigma^2$**  We first trained VAEs with 200 latent dimensions and fixed values of $\sigma^2$ for the observation model. Given our previous analysis, this is flawed but unfortunately is representative of a wide range of open source implementations we surveyed. Moreover, this setting allows us to better understand how KL-annealing (Bowman et al., 2015; Sønderby et al., 2016) impacts posterior collapse.

Figure 3 shows the ELBO during training of an MNIST VAE with 2 hidden layers in both the encoder and decoder, and a stochastic layer with 200 hidden units. Figure 4 shows the cumulative distribution of the per-dimension KL divergence between the variational distribution and the prior at the end of training. We observe that using a smaller value of $\sigma^2$ prevents the posterior from collapsing and allows the model to achieve a substantially higher ELBO. It is possible that the difference in ELBO is due entirely to the change of scale introduced by $\sigma^2$ and not because of differences in the learned representations. To test this hypothesis we took each of the trained models and optimized for $\sigma^2$ while keeping all other parameters fixed (Table 1). As expected, the ELBO increased but the relative ordering remained the same with a significant gap still present.

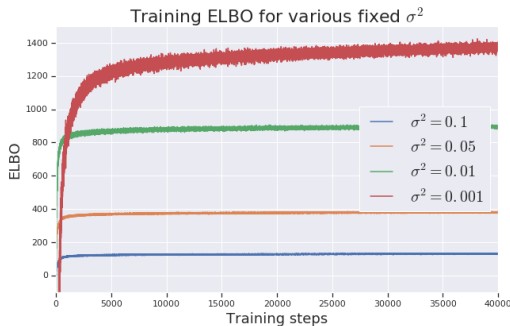

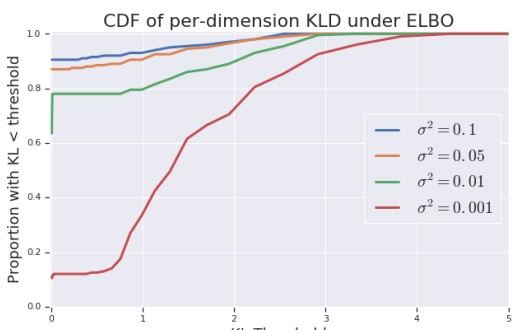

Figure 3: ELBO during training of MNIST VAEs with Gaussian observation model. A better ELBO is achieved with a smaller choice of $\sigma^2$.

Figure 4: The proportion of inactive units in trained MNIST VAEs which, on average, are less than the specified threshold.

| Model | ELBO | $\sigma^2$-tuned ELBO |
|---|---|---|
| $\sigma^2 = 0.1$ | 130.3 | 1302.9 |
| $\sigma^2 = 0.05$ | 378.7 | 1376.0 |
| $\sigma^2 = 0.01$ | 893.6 | 1435.1 |
| $\sigma^2 = 0.001$ | 1379.0 | 1485.9 |

Table 1: Evaluation of trained MNIST VAEs. The final model is evaluated on the training set. We also tuned $\sigma^2$ to the trained model and re-evaluated to confirm that the difference in loss is due to differences in latent representations.

Figure 5: Proportion of inactive units thresholded by KL divergence when using 0-1 KL-annealing. The solid line represents the final model while the dashed line is the model after only 80 epochs of training.

**The role of KL-annealing** An alternative approach to tuning $\sigma^2$ is to scale the KL term directly by a coefficient, $\beta$. For $\beta < 1$ this provides a loose lowerbound on the ELBO but for appropriate choices of $\beta$ and learning rate, this scheme can be made equivalent to tuning $\sigma^2$. In this section we explore this technique. We found that KL-annealing may provide temporary relief from posterior collapse but that if $\sigma^2$ is not appropriately tuned then ultimately ELBO training will recover the default solution. In Figure 5 we show the proportion of units collapsed by threshold for several fixed choices of $\sigma^2$ when $\beta$ is annealed from 0 to 1 over the first 100 epochs. The solid lines correspond to the final model while the dashed line corresponds to the model at 80 epochs of training. Early on, KL-annealing is able to reduce posterior collapse but ultimately we recover the ELBO solution from Figure 4.

After finding that KL-annealing alone was insufficient to prevent posterior collapse we explored KL annealing while learning $\sigma^2$. Based on our analysis in the linear case we expect that this should work well: while $\beta$ is small the model should be able to learn to reduce $\sigma^2$. To test this, we trained the same VAE as above on MNIST data but this time we allowed $\sigma^2$ to be learned. The results are presented in Figure 6. We trained first using the standard ELBO objective and then again using KL-annealing. The ELBO objective learns to reduce $\sigma^2$ but ultimately learns a solution with a large degree of posterior collapse. Using KL-annealing, the VAE is able to learn a much smaller $\sigma^2$ value and ultimately reduces posterior collapse. Interestingly, despite significantly differing representations, these two models have approximately the same final training ELBO. This is consistent with the analysis of Alemi et al. (2017), who showed that there can exist solutions equal under ELBO with differing posterior distributions.

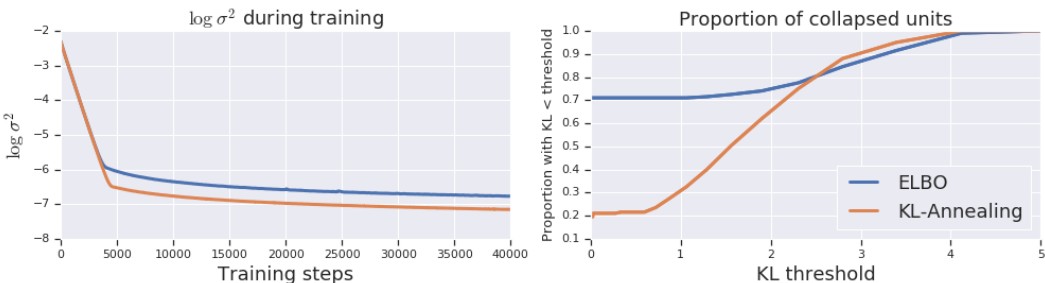

Figure 6: Comparing learned solutions using KL-Annealing versus standard ELBO training when $\sigma^2$ is learned.

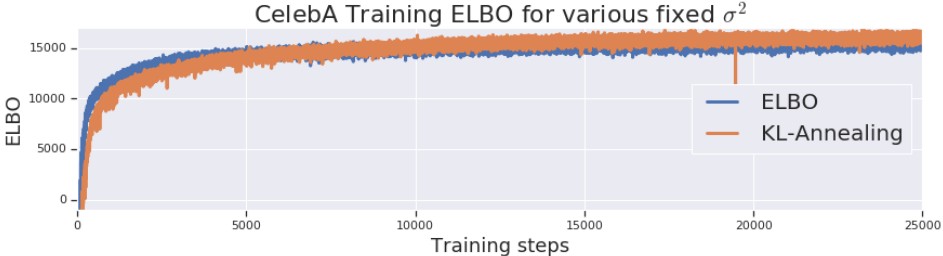

Figure 7: ELBO during training of convolutional CelebA VAEs while learning $\sigma^2$.

### 5.2.1 OTHER DATASETS

We trained deep convolutional VAEs with 500 hidden dimensions on images from the CelebA dataset (resized to 64x64). In Figure 7 we show the training ELBO for the standard ELBO objective and training with KL-annealing. In each case, $\sigma^2$ is learned online. As in Figure 6, KL-Annealing enabled the VAE to learn a smaller value of $\sigma^2$ which corresponded to a better final ELBO value and reduced posterior collapse (Figure 8).

## 6 DISCUSSION

By analyzing the correspondence between linear VAEs and pPCA we have made significant progress towards understanding the causes of posterior collapse. We have shown that for simple linear VAEs posterior collapse is caused by spurious local maxima in the marginal log-likelihood and we demonstrated empirically that the same local maxima seem to play a role when optimizing deep non-linear VAEs. In future work, we hope to extend this analysis to other observation models and provide theoretical support for the non-linear case.

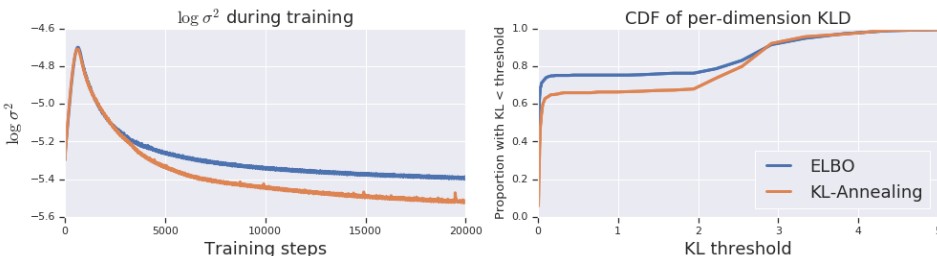

Figure 8: Learning $\sigma^2$ for CelebA VAEs with standard ELBO training and KL-Annealing. KL-Annealing enables a smaller $\sigma^2$ to be learned and reduces posterior collapse.

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

## A  STATIONARY POINTS OF PPCA

Here we briefly summarize the analysis of (Tipping & Bishop, 1999) with some simple additional observations. We recommend that interested readers study Appendix A of Tipping & Bishop (1999) for the full details. We begin by formulating the conditions for stationary points of $\sum_{\mathbf{x}_i} \log p(\mathbf{x}_i)$:

$$\mathbf{SC}^{-1}\mathbf{W} = \mathbf{W} \tag{10}$$

Where $\mathbf{S}$ denotes the sample covariance matrix (assuming we set $\boldsymbol{\mu} = \boldsymbol{\mu}_{MLE}$, *which we do throughout*), and $\mathbf{C} = \mathbf{WW}^T + \sigma^2 I$ (note that the dimensionality is different to $\mathbf{M}$). There are three possible solutions to this equation, (1) $\mathbf{W} = \mathbf{0}$, (2) $\mathbf{C} = \mathbf{S}$, or (3) the more general solutions. (1) and (2) are not particularly interesting to us, so we focus herein on (3).

We can write $\mathbf{W} = \mathbf{ULV}^T$ using its singular value decomposition. Substituting back into the stationary points equation, we recover the following:

$$\mathbf{SUL} = \mathbf{U}(\sigma^2 I + \mathbf{L}^2)\mathbf{L} \tag{11}$$

Noting that $\mathbf{L}$ is diagonal, if the $j^{th}$ singular value ($l_j$) is non-zero, this gives $\mathbf{Su}_j = (\sigma^2 + l_j^2)\mathbf{u}_j$, where $u_j$ is the $j^{th}$ column of $\mathbf{U}$. Thus, $\mathbf{u}_j$ is an eigenvector of $\mathbf{S}$ with eigenvalue $\lambda_j = \sigma^2 + l_j^2$. For $l_j = 0$, $\mathbf{u}_j$ is arbitrary.

Thus, all potential solutions can be written as, $\mathbf{W} = U_q(K_q - \sigma^2 I)^{1/2}\mathbf{R}$, with singular values written as $k_j = \sigma^2$ or $\sigma^2 + l_j^2$ and with $\mathbf{R}$ representing an arbitrary orthogonal matrix.

From this formulation, one can show that the global optimum is attained with $\sigma^2 = \sigma^2_{MLE}$ and $U_q$ and $K_q$ chosen to match the leading singular vectors and values of $\mathbf{S}$.

### A.1  STABILITY OF STATIONARY POINT SOLUTIONS

Consider stationary points of the form, $\mathbf{W} = \mathbf{U}_q(K_q - \sigma^2 I)^{1/2}$ where $\mathbf{U}_q$ contains arbitrary eigenvectors of $\mathbf{S}$. In the original pPCA paper they show that all solutions except the leading principal components correspond to saddle points in the optimization landscape. However, this analysis depends critically on $\sigma^2$ being set to the true maximum likelihood estimate. Here we repeat their analysis, considering other (fixed) values of $\sigma^2$.

We consider a small perturbation to a column of $\mathbf{W}$, of the form $\epsilon\mathbf{u}_j$ . To analyze the stability of the perturbed solution, we check the sign of the dot-product of the perturbation with the likelihood gradient at $\mathbf{w}_i + \epsilon\mathbf{u}_j$. Ignoring terms in $\epsilon^2$ we can write the dot-product as,

$$\epsilon N(\lambda_j/k_i - 1)\mathbf{u}_j^T\mathbf{C}^{-1}\mathbf{u}_j \tag{12}$$

Now, $\mathbf{C}^{-1}$ is positive definite and so the sign depends only on $\lambda_j/k_i - 1$. The stationary point is stable (local maxima) only if the sign is negative. If $k_i = \lambda_i$ then the maxima is stable only when $\lambda_i > \lambda_j$, in words, the top $q$ principal components are stable. However, we must also consider the case $k = \sigma^2$. Tipping & Bishop (1999) show that if $\sigma^2 = \sigma^2_{MLE}$, then this also corresponds to a saddle point as $\sigma^2$ is the average of the smallest eigenvalues meaning some perturbation will be unstable (except in a special case which is handled separately).

However, what happens if $\sigma^2$ is not set to be the maximum likelihood estimate? In this case, it is possible that there are no unstable perturbation directions (that is, $\lambda_j < \sigma^2$ for too many $j$). In this case when $\sigma^2$ is fixed, there are local optima where $\mathbf{W}$ has zero-columns — the same solutions that we observe in non-linear VAEs corresponding to posterior collapse. Note that when $\sigma^2$ is learned in non-degenerate cases the local maxima presented above become saddle points where $\sigma^2$ is made smaller by its gradient. In practice, we find that even when $\sigma^2$ is learned in the non-linear case local maxima exist.

# B    Identifiability of the linear VAE

Linear autoencoders suffer from a lack of identifiability which causes the decoder columns to span the principal component subspace instead of recovering it. Here we show that linear VAEs are able to recover the principal components up to scaling.

We once again consider the linear VAE from Eq. (9):

$$p(\mathbf{x} \mid \mathbf{z}) = \mathcal{N}(\mathbf{Wz} + \boldsymbol{\mu}, \sigma^2 \mathbf{I}),$$
$$q(\mathbf{z} \mid \mathbf{x}) = \mathcal{N}(\mathbf{V}(\mathbf{x} - \boldsymbol{\mu}), \mathbf{D}),$$

The output of the VAE, $\tilde{\mathbf{x}}$ is distributed as,

$$\tilde{\mathbf{x}}|\mathbf{x} \sim \mathcal{N}(\mathbf{WV}(\mathbf{x} - \boldsymbol{\mu}) + \boldsymbol{\mu}, \sigma^2 \mathbf{W}\mathbf{D}^{-1}\mathbf{W}^T).$$

Therefore, the linear VAE is invariant to the following transformation:

$$\mathbf{W} \leftarrow \mathbf{WA},$$
$$\mathbf{V} \leftarrow \mathbf{A}^{-1}\mathbf{V}, \tag{13}$$
$$\mathbf{D} \leftarrow \mathbf{A}^{-1}\mathbf{DA}^{-1},$$

where $\mathbf{A}$ is a diagonal matrix with non-zero entries so that $\mathbf{D}$ is well-defined. We see that the direction of the columns of $\mathbf{W}$ are always identifiable, and thus the principal components can be exactly recovered.

Moreover, we can recover complete identifiability by fixing $\mathbf{D} = \mathbf{I}$, so that there is a unique global maximum.

# C    Stationary points of ELBO

Here we present details on the analysis of the stationary points of the ELBO objective. To begin, we first derive closed form solutions to the components of the marginal log-likelihood (including the ELBO). The VAE we focus on is the one presented in Eq. (9), with a linear encoder, linear decoder, Gaussian prior, and Gaussian observation model.

Remember that one can express the marginal log-likelihood as:

$$\log p(\mathbf{x}) = \overset{(A)}{KL(q(\mathbf{z}|\mathbf{x})||p(\mathbf{z}|\mathbf{x}))} - \overset{(B)}{KL(q(\mathbf{z}|\mathbf{x})||p(z))} + \overset{(C)}{\mathbb{E}_{q(\mathbf{z}|\mathbf{x})}[\log p(\mathbf{x}|\mathbf{z})]}. \tag{14}$$

Each of the terms (A-C) can be expressed in closed form for the linear VAE. Note that the KL term (A) is minimized when the variational distribution is exactly the true posterior distribution. This is possible when the columns of the decoder are orthogonal.

The term (B) can be expressed as,

$$KL(q(\mathbf{z}|\mathbf{x})||p(z)) = 0.5(-\log\det\mathbf{D} + (\mathbf{x} - \boldsymbol{\mu})^T\mathbf{V}^T\mathbf{V}(\mathbf{x} - \boldsymbol{\mu}) + tr(\mathbf{D}) - q). \tag{15}$$

The term (C) can be expressed as,

$$\mathbb{E}_{q(\mathbf{z}|\mathbf{x})}[\log p(\mathbf{x}|\mathbf{z})] = \mathbb{E}_{q(\mathbf{z}|\mathbf{x})}\left[-(\mathbf{Wz} - (\mathbf{x} - \boldsymbol{\mu}))^T(\mathbf{Wz} - (\mathbf{x} - \boldsymbol{\mu}))/2\sigma^2 - \frac{d}{2}\log 2\pi\sigma^2\right] \tag{16}$$

$$= \mathbb{E}_{q(\mathbf{z}|\mathbf{x})}\left[\frac{-(\mathbf{Wz})^T(\mathbf{Wz}) + 2(\mathbf{x} - \boldsymbol{\mu})^T\mathbf{Wz} - (\mathbf{x} - \boldsymbol{\mu})^T(\mathbf{x} - \boldsymbol{\mu})}{2\sigma^2} - \frac{d}{2}\log 2\pi\sigma^2\right].$$
$$\tag{17}$$

Noting that $\mathbf{Wz} \sim \mathcal{N}\left(\mathbf{WV}(\mathbf{x} - \boldsymbol{\mu}), \mathbf{WDW}^T\right)$, we can compute the expectation analytically and obtain,

$$\mathbb{E}_{q(\mathbf{z}|\mathbf{x})}\left[\log p(\mathbf{x}|\mathbf{z})\right] = \frac{1}{2\sigma^2}[-tr(\mathbf{WDW}^T) - (\mathbf{x} - \boldsymbol{\mu})^T\mathbf{V}^T\mathbf{W}^T\mathbf{WV}(\mathbf{x} - \boldsymbol{\mu}) \tag{18}$$

$$+ 2(\mathbf{x} - \boldsymbol{\mu})^T\mathbf{WV}(\mathbf{x} - \boldsymbol{\mu}) - (\mathbf{x} - \boldsymbol{\mu})^T(\mathbf{x} - \boldsymbol{\mu})] - \frac{d}{2}\log 2\pi\sigma^2. \tag{19}$$

To compute the stationary points we must take derivatives with respect to $\boldsymbol{\mu}, \mathbf{D}, \mathbf{W}, \mathbf{V}, \sigma^2$. As before, we have $\boldsymbol{\mu} = \boldsymbol{\mu}_{MLE}$ at the global maximum and for simplicity we fix $\boldsymbol{\mu}$ here for the remainder of the analysis.

Taking the marginal likelihood over the whole dataset, at the stationary points we have,

$$\frac{\partial}{\partial \mathbf{D}}(-(B) + (C)) = \frac{N}{2}(\mathbf{D}^{-1} - \mathbf{I} - \frac{1}{\sigma^2}\text{diag}(\mathbf{W}^T\mathbf{W})) = 0 \tag{20}$$

$$\frac{\partial}{\partial \mathbf{V}}(-(B) + (C)) = \frac{N}{\sigma^2}(\mathbf{W}^T - (\mathbf{W}^T\mathbf{W} + \sigma^2\mathbf{I})\mathbf{V})\mathbf{S} = 0 \tag{21}$$

$$\frac{\partial}{\partial \mathbf{W}}(-(B) + (C)) = \frac{N}{\sigma^2}(\mathbf{SV}^T - \mathbf{DW} - \mathbf{WVSV}^T) = 0 \tag{22}$$

The above are computed using standard matrix derivative identities (Petersen et al.). These equations yield the expected solution for the variational distribution directly. From Eq. (20) we compute $\mathbf{D}^* = \sigma^2(\text{diag}(\mathbf{W}^T\mathbf{W}) + \sigma^2\mathbf{I})^{-1}$ and $\mathbf{V}^* = \mathbf{M}^{-1}\mathbf{W}^T$, recovering the true posterior mean in all cases and getting the correct posterior covariance when the columns of $\mathbf{W}$ are orthogonal. We will now proceed with the proof of Theorem 1.

**Theorem 1.** *The ELBO objective does not introduce any additional local maxima to the pPCA model.*

*Proof.* If the columns of $\mathbf{W}$ are orthogonal then the marginal log-likelihood is recovered exactly at all stationary points. This is a direct consequence of the posterior mean *and* covariance being recovered exactly at all stationary points so that (1) is zero.

We must give separate treatment to the case where there is a stationary point without orthogonal columns of $\mathbf{W}$. Suppose we have such a stationary point, using the singular value decomposition we can write $\mathbf{W} = \mathbf{ULR}^T$, where $\mathbf{U}$ and $\mathbf{R}$ are orthogonal matrices. Note that $\log p(\mathbf{x})$ is invariant to the choice of $\mathbf{R}$ (Tipping & Bishop, 1999). However, the choice of $\mathbf{R}$ does have an effect on the first term (1) of Eq. (14): this term is minimized when $\mathbf{R} = \mathbf{I}$, and thus the ELBO must increase.

To formalize this argument, we compute (1) at a stationary point. From above, at every stationary point the mean of the variational distribution exactly matches the true posterior. Thus the KL simplifies to:

$$KL(q(\mathbf{z}|\mathbf{x})||p(\mathbf{z}|\mathbf{x})) = \frac{1}{2}\left(tr(\frac{1}{\sigma^2}\mathbf{MD}) - q + q\log\sigma^2 - \log(\det\mathbf{M}\det\mathbf{D})\right), \tag{23}$$

$$= \frac{1}{2}\left(tr(\mathbf{M}\widetilde{\mathbf{M}}^{-1}) - q - \log\frac{\det\mathbf{M}}{\det\widetilde{\mathbf{M}}}\right), \tag{24}$$

$$= \frac{1}{2}\left(\sum_{i=1}^{q}\frac{\mathbf{M}_{ii}}{\widetilde{\mathbf{M}}_{ii}} - q - \log\det\mathbf{M} + \log\det\widetilde{\mathbf{M}}\right), \tag{25}$$

$$= \frac{1}{2}\left(\log\det\widetilde{\mathbf{M}} - \log\det\mathbf{M}\right), \tag{26}$$

$$\tag{27}$$

where $\widetilde{\mathbf{M}} = \text{diag}(\mathbf{W}^T\mathbf{W}) + \sigma^2\mathbf{I}$. Now consider applying a small rotation to $\mathbf{W}$: $\mathbf{W} \mapsto \mathbf{WR}_\epsilon$. As the optimal $\mathbf{D}$ and $\mathbf{V}$ are continuous functions of $\mathbf{W}$, this corresponds to a small

perturbation of these parameters too for a sufficiently small rotation. Importantly, $\log \det \mathbf{M}$ remains fixed for any orthogonal choice of $\mathbf{R}_\epsilon$ but $\log \det \widetilde{\mathbf{M}}$ does not. Thus, we choose $\mathbf{R}_\epsilon$ to minimize this term. In this manner, (1) shrinks meaning that the ELBO (-2)+(3) must increase. Thus if the stationary point existed, it must have been a saddle point.

We now describe how to construct such a small rotation matrix. First note that without loss of generality we can assume that $\det(\mathbf{R}) = 1$. (Otherwise, we can flip the sign of a column of $\mathbf{R}$ and the corresponding column of $\mathbf{U}$.) And additionally, we have $\mathbf{WR} = \mathbf{UL}$, which is orthogonal.

The Special Orthogonal group of determinant 1 orthogonal matrices is a compact, connected Lie group and therefore the exponential map from its Lie algebra is surjective. This means that we can find an upper-triangular matrix $\mathbf{B}$, such that $\mathbf{R} = \exp\{\mathbf{B} - \mathbf{B}^T\}$. Consider $\mathbf{R}_\epsilon = \exp\{\frac{1}{n(\epsilon)}(\mathbf{B} - \mathbf{B}^T)\}$, where $n(\epsilon)$ is an integer chosen to ensure that the elements of $\mathbf{B}$ are within $\epsilon > 0$ of zero. This matrix is a rotation in the direction of $\mathbf{R}$ which we can make arbitrarily close to the identity by a suitable choice of $\epsilon$. This is verified through the Taylor series expansion of $\mathbf{R}_\epsilon = I + \frac{1}{n(\epsilon)}(\mathbf{B} - \mathbf{B}^T) + O(\epsilon^2)$. Thus, we have identified a small perturbation to $\mathbf{W}$ (and $\mathbf{D}$ and $\mathbf{V}$) which decreases the posterior KL (A) but keeps the marginal log-likelihood constant. Thus, the ELBO increases and the stationary point must be a saddle point.

$\square$

## C.1 Bernoulli Probabilistic PCA

We would like to extend our linear analysis to the case where we have a Bernoulli observation model, as this setting also suffers severely from posterior collapse. The analysis may also shed light on more general categorical observation models which have also been used. Typically, in these settings a continuous latent space is still used (for example, Bowman et al. (2015)).

We will consider the following model,

$$
\begin{aligned}
p(\mathbf{z}) &= \mathcal{N}(0, \mathbf{I}), \\
p(\mathbf{x}|\mathbf{z}) &= \text{Bernoulli}(\mathbf{y}), \\
\mathbf{y} &= \sigma(\mathbf{Wz} + \boldsymbol{\mu})
\end{aligned}
\tag{28}
$$

where $\sigma$ denotes the sigmoid function, $\sigma(y) = 1/(1+\exp(-y))$ and we assume an independent Bernoulli observation model over $\mathbf{x}$.

Unfortunately, under this model it is difficult to reason about the stationary points. There is no closed form solution for the marginal likelihood $p(\mathbf{x})$ or the posterior distribution $p(\mathbf{z}|\mathbf{x})$. Numerical integration methods exist which may make it easy to evaluate this quantity in practice but they will not immediately provide us a good gradient signal.

We can compute the density function for $\mathbf{y}$ using the change of variables formula. Noting that $\mathbf{Wz} + \boldsymbol{\mu} \sim \mathcal{N}(\boldsymbol{\mu}, \mathbf{WW}^T)$, we recover the following logit-Normal distribution:

$$
f(\mathbf{y}) = \frac{1}{\sqrt{2\pi|\mathbf{WW}^T|}} \frac{1}{\Pi_i y_i(1-y_i)} \exp\{-\frac{1}{2}\left(\log(\frac{\mathbf{y}}{1-\mathbf{y}}) - \boldsymbol{\mu}\right)^T (\mathbf{WW}^T)^{-1}\left(\log(\frac{\mathbf{y}}{1-\mathbf{y}}) - \boldsymbol{\mu}\right)\}
\tag{29}
$$

We can write the marginal likelihood as,

$$
p(\mathbf{x}) = \int p(\mathbf{x}|\mathbf{z})p(\mathbf{z})d\mathbf{z},
\tag{30}
$$

$$
= \mathbb{E}_{\mathbf{z}}\left[\mathbf{y}(\mathbf{z})^{\mathbf{x}}(1 - \mathbf{y}(\mathbf{z}))^{1-\mathbf{x}}\right],
\tag{31}
$$

where $(\cdot)^{\mathbf{x}}$ is taken to be elementwise. Unfortunately, the expectation of a logit-normal distribution has no closed form (Atchison & Shen, 1980) and so we cannot tractably compute the marginal likelihood.

Similarly, under ELBO we need to compute the expected reconstruction error. This can be written as,

$$\mathbb{E}_{q(\mathbf{z}|\mathbf{x})}[\log p(\mathbf{x}|\mathbf{z})] = \int \mathbf{y}(\mathbf{z})^{\mathbf{x}}(1 - \mathbf{y}(\mathbf{z}))^{1-\mathbf{x}}\mathcal{N}(\mathbf{z}; \mathbf{V}(\mathbf{x} - \boldsymbol{\mu}), \mathbf{D})d\mathbf{z}, \tag{32}$$

another intractable integral.

## D    EXPERIMENT DETAILS

**Visualizing stationary points of pPCA**    For this experiment we computed the pPCA MLE using a subset of 10000 random training images from the MNIST dataset. We evaluate and plot the marginal log-likelihood in closed form on this same subset.

**MNIST VAE**    The VAEs we trained on MNIST all had the same architecture: 784-1024-512-k-512-1024-784. The VAE parameters were optimized jointly using the Adam optimizer (Kingma & Ba, 2014). We trained the VAE for 1000 epochs total, keeping the learning rate fixed throughout. We performed a grid search over values for the learning rate and reported results for the model which achieved the best training ELBO.

**CelebA VAE**    We used the convolutional architecture proposed by Higgins et al. (2016). Otherwise, the experimental procedure followed that of the MNIST VAEs.

### D.1    ADDITIONAL RESULTS

We also trained convolutional VAEs on the CelebA dataset using fixed choices of $\sigma^2$. As expected, the same general pattern emerged as in Figure 3.

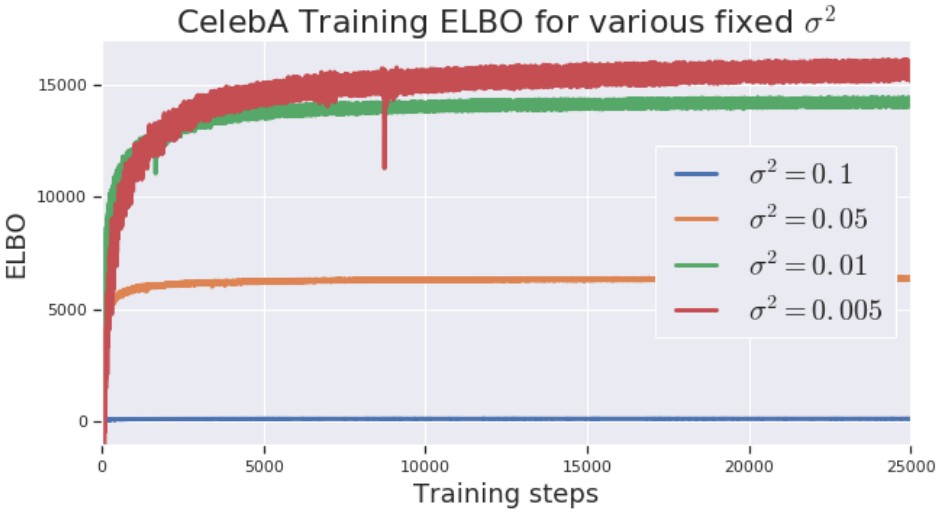

Figure 9: ELBO during training of convolutional CelebA VAEs with fixed $\sigma^2$.

Reconstructions from the KL-Annealed model are shown in Figure 10. We also show the output of interpolating in the latent space in Figure 11. To produce the latter plot, we compute the variational mean of 3 input points (top left, top right, bottom left) and interpolate linearly between. We also extrapolate out to a fourth point (bottom right), which lies on the plane defined by the other points.

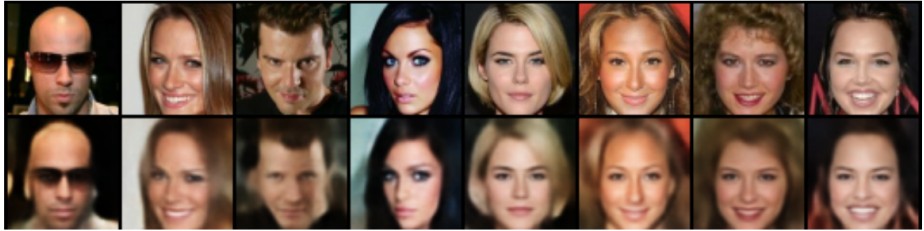

Figure 10: Reconstructions from the convolutional VAE trained with KL-Annealing on CelebA.

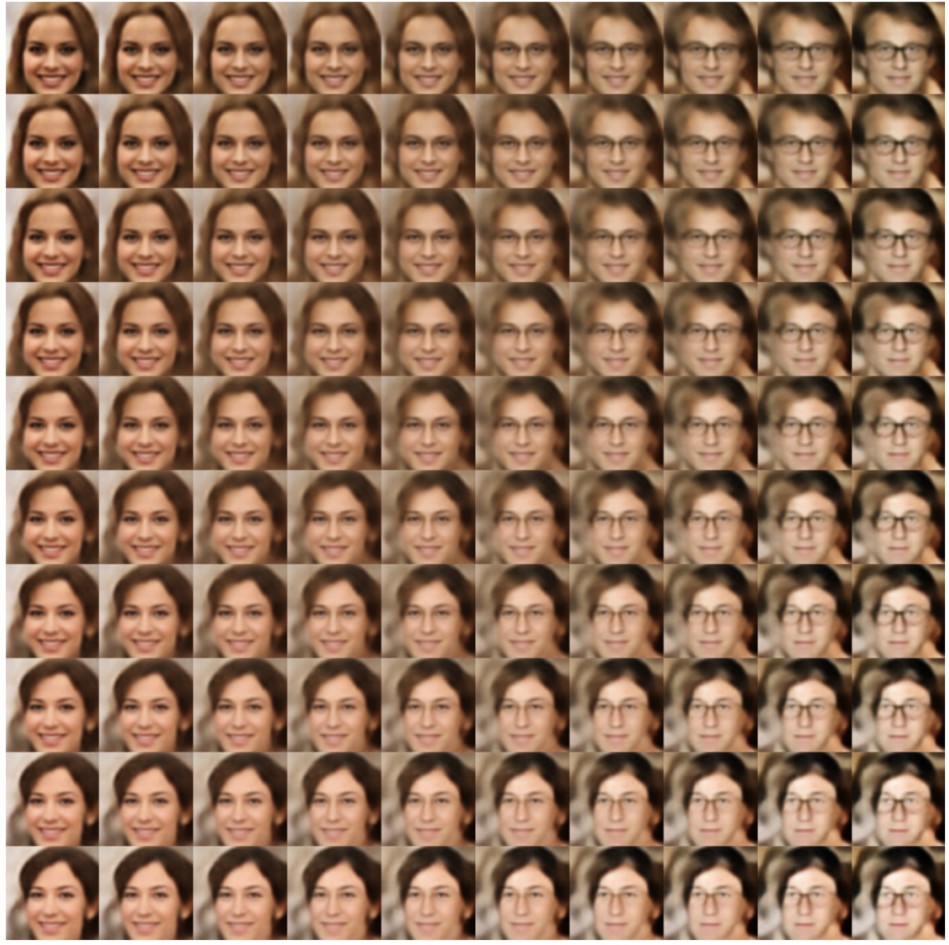

Figure 11: Latent space interpolations from the convolutional VAE trained with KL-Annealing on CelebA.

