# OpenReview forum: "Understanding Posterior Collapse in Generative Latent Variable Models"
_ICLR.cc/2019/Workshop/DeepGenStruct — DeepGenStruct 2019_

### Official Review · AnonReviewer1 · 2019-04-10
**The paper draws connections between pPCA and the linear VAE in order to tackle the mode collapse problem. The connections between pPCA and linear VAEs are known**

**Rating:** 3
**Confidence:** 3

**Review:**

This paper draws connections between pPCA and linear VAEs. I would like to argue that part is straighforward, due to the following facts:
1. The exact pPCA posterior is a Gaussian whose mean depends linearly on x.
2. The variational family is also linear on x; therefore it includes the exact posterior.
3. Variational inference minimizes the KL between the variational family and the exact posterior.
4. The expectations in the ELBO can be analytically computed.
Given these points, it is easy to conclude that variational inference will find the variational distribution that matches the posterior exactly, therefore recovering pPCA.

The rest of the paper uses the insights from the first part to analyze mode collapse in non-linear VAEs. I found that part more interesting than the former.

---

### Official Review · AnonReviewer2 · 2019-04-15
**a different perspective in understanding posterior collapse in VAEs**

**Rating:** 4
**Confidence:** 2

**Review:**

This paper presents a different perspective in understanding posterior collapse in VAEs by studying the connection between probabilistic PCA (pPCA) and linear VAEs. In previous work, it is widely acknowledged that the KL-term plays an important role in posterior collapse. However, in this paper, the authors show theoretically that even the marginal log-likelihood itself could have spurious local optima and for linear VAEs, the ELBO does not add any additional optima to the pPCA model. Experimental results on posterior collapse in non-linear VAEs provide evidence to the analysis.

The analysis in this paper presents a complementary view in helping us better understand posterior collapse. One thing which is not clear to me in the current analysis is how posterior collapse can happen to pPCA if it has the same spurious local optima. Another minor comment is on the MNIST dataset, given the data is practically binary, I wonder if that would also contribute to the preference of having a smaller \sigma. Finally, it would be interesting if the authors could also comment on VAEs with other observational model rather than Gaussian (e.g., Multinomial, as in Krishnan et al. 2018, On the challenges of learning with inference networks on sparse, high-dimensional data).

---

### Decision · Program_Chairs · 2019-04-19
**Acceptance Decision**

Accept